# Improved YOLOv5 Network for Real-Time Object Detection in Vehicle-Mounted Camera Capture Scenarios

**DOI:** 10.3390/s23104589

**Published:** 2023-05-09

**Authors:** Zuyue Ren, Hong Zhang, Zan Li

**Affiliations:** School of Information Engineering, Minzu University of China, Beijing 100080, China

**Keywords:** road crack detection, traffic sign detection, YOLO neural network, Bi-FPN, CBAM attention model

## Abstract

Object detection in the process of driving is a convenient and efficient task. However, due to the complex transformation of the road environment and vehicle speed, the scale of the target will not only change significantly but also be accompanied by the phenomenon of motion blur, which will have a significant impact on the detection accuracy. In practical application scenarios, it is difficult for traditional methods to simultaneously take into account the need for real-time detection and high accuracy. To address the above problems, this study proposes an improved network based on YOLOv5, taking traffic signs and road cracks as detection objects and conducting separate research. This paper proposes a GS-FPN structure to replace the original feature fusion structure for road cracks. This structure integrates the convolutional block attention model (CBAM) based on bidirectional feature pyramid networks (Bi-FPN) and introduces a new lightweight convolution module (GSConv) to reduce the information loss of the feature map, enhance the expressive ability of the network, and ultimately achieve improved recognition performance. For traffic signs, a four-scale feature detection structure is used to increase the detection scale of shallow layers and improve the recognition accuracy for small targets. In addition, this study has combined various data augmentation methods to improve the robustness of the network. Through experiments using 2164 road crack datasets and 8146 traffic sign datasets made by LabelImg, compared to the baseline model (YOLOv5s), the modified YOLOv5 network improves the mean average precision (mAP) result of the road crack dataset and small targets in the traffic sign dataset by 3% and 12.2%, respectively.

## 1. Introduction

Road cracks and traffic signs are two essential factors in high-speed transportation systems. Both are not only closely related to road safety [1,2], but also an important basis for realizing the concept of autonomous driving. Therefore, achieving efficient, accurate detection of traffic signs and road cracks is the key issue in formulating driverless technology [3] development direction and road maintenance decision-making. With the continuous progress in the field of deep learning, detection methods based on neural networks have gradually been applied to the industrial field. Examples include R-CNN, Fast-RCNN, Fast-RCNN, Mask-RCNN, Alex Net [4,5,6,7,8], and YOLO [9,10,11,12]. These models have achieved good results in detecting corresponding targets under ideal conditions. However, there are still some problems in the actual application scenarios under natural and realistic conditions of high-speed driving: (1) the model is too large; (2) poor real-time performance; (3) the accuracy of a single scale is low. Traditional CNN models require large numbers of parameters and floating point operations per second (FLOPs), leading to the limited resources of mobile devices that cannot complete excessive network deployment. In addition, due to the setting of the candidate area, the two-stage model sometimes cannot meet the real-time detection requirements of high-speed driving [13]. Therefore, in order to meet the two prerequisites of real-time performance and model size while maintaining accuracy, this study selected the one-stage detection algorithm YOLOv5, which requires less computation and is fast.

Considering that the features of road cracks and traffic signs differ greatly in the shape, color, and size of the target in the image, this study generates two datasets for research. Compared with traffic signs, road cracks are more subtle, so feature extraction needs to be softer to avoid excessive redundant information drowning the target features. The shape and color features of road signs are more obvious, and the standard convolution kernel can extract the main features more effectively. At the same time, as the camera moved, it was found that the size of traffic signs and road cracks in the image changed to different degrees. Considering the above problems, we finally designed two different networks to achieve the best effect of corresponding target recognition.

This study’s main contributions to road crack detection are summarized in three parts as follows:The CBAM has been added to the backbone network, which contains channel relationships and spatial locations that help refine target features and improve the ability to extract target features.The Bi-FPN is used to replace the original FPN structure of YOLOv5 to enhance the effect of multi-scale feature fusion, by using the adaptive weight to distinguish the importance of feature maps from different layers, enhancing important features and inhibiting features that are not significant.The convolution module of the neck layer is replaced with a lightweight convolution module, named GSConv, which can significantly reduce model parameters and computational complexity while maintaining model accuracy.

This study’s main contributions to traffic sign detection are summarized in two parts as follows:Using a four-scale feature detection structure, a large-scale detector head is added to detect small targets, ensuring the detection rate of large and medium-sized targets while improving the detection effect for small targets.In the face of the problem of multiple types of traffic signs and fewer samples for each category, various data augmentation methods have been combined to enrich training samples, improve the robustness of the model, and avoid overfitting problems.

The rest of this paper is organized as follows: Section 2 introduces the related work and analysis of road crack and traffic sign detection based on CNN, as well as the original YOLOv5 model. Section 3 and Section 4 introduce the methods of efficient real-time detection of road cracks and traffic signs, respectively, and the improvement of the network. The experimental configuration, dataset, experimental results, and analysis are explained in Section 5. Finally, the Section 6 is the conclusion of this paper.

## 2. Related Work

### 2.1. Existing Work

Early research on object detection is mostly based on image processing algorithms, which extract and classify objects according to their color, shape, edge, and other features [14]. Liu et al. [15] proposed a binary classification method for concrete crack recognition based on crack features. However, the classification threshold of this method needs to be set in advance, which has certain limitations and cannot adapt to complex road crack scenarios. Wang et al. [16] used the orthogonal projection method to define and evaluate the dataset and finally divided the crack image into three levels. The proposed method has a good detection effect and robustness. Noh et al. [17] proposed an automatic crack detection and segmentation method for concrete images based on fuzzy C-means clustering and multiple noise reduction, which can effectively distinguish between background and cracks. However, when the crack location contains a lot of noise, the detection effect is not so successful. In the face of high-speed driving scenes, image noise is inevitable. At the same time, if the hardware performance is improved, the cost is too high, and this method is not suitable for real industrial scenarios. Although early image processing techniques can also help complete certain detection tasks, these methods can achieve good results under ideal conditions, but in the actual application of industrial scenes, compared with neural networks, they are still unsatisfactory.

With the development of convolutional neural networks, neural network models are more and more widely used in object detection. Song et al. [18] proposed a crack CNN based on the combination of a CNN and the Beamlet algorithm to detect cracks, effectively overcoming the problem of crack noise interference and improving the accuracy of crack detection. Xu et al. [19] proposed an improved network based on Faster R-CNN. This network uses a region proposal network (RPN) to identify rectangular boundary boxes through shared convolution features and classify and locate multiple targets. Pena-Caballero et al. [20] used the YOLO9000 network and semantic segmentation to classify and detect pixel-level objects. However, because of the use of image segmentation technology, it is easily affected by the noise produced by high-speed motion. Soetedjo et al. [21] detect traffic signs using a method combining maximum stable outer region (MSER) and Lucas Kanade tracking (LKT). A feature template matching technique based on a gradient histogram was used to verify the candidate objects and reduce false positives. Tong et al. [22] introduced a pilot module based on the YOLOv2 model to expand the depth and width of the network at extra cost. At the same time, the loss function of the network was modified to improve the influence of small targets on the network and optimize the detection results.

All of these networks are accomplished under ideal conditions. In the face of complex application scenarios, real-time performance cannot be guaranteed, and the network is subjected to environmental interference. Meanwhile, multi-scale target detection is ignored. As the distance between the camera and the target decreases, the size of the target in the image will also change. For neural networks, the visual features of large and small objects are completely different. With the deepening of the network layer, the feature information and location information of small targets are gradually lost [23]. In the face of complex highways, the detection capability of a single-scale network will be reduced.

Faced with this challenge, the multi-scale feature fusion method can combine the feature information of a shallow network with the location information of a deep network to improve the detection ability [24]. The feature fusion structure of the feature pyramid network (FPN) [25] adopted by the neck layer of YOLOv5 is one of the most popular multi-scale networks. Liu et al. [26] proposed a multi-scale regional convolutional neural network to conduct multi-scale deconvolution post-sampling of deep convolutional features, splice them with shallow features, and finally construct new feature mappings. The Bi-FPN structure proposed by Tan et al. [27] is an improved feature fusion method based on the FPN structure, which has been widely used since it was proposed. In the study of deep supervised convolutional neural networks, Qu et al. [28] tried to introduce deep features into shallow features at different stages of convolution. Wang et al. [29] proposed an AF-PAN structure to enhance feature extraction in feature fusion networks by introducing an adaptive attention module (AAM) and a feature enhancement module (FEM).

Inspired by these findings, this study designed a simple and effective feature fusion structure named GS-BiFPN in the study of road cracks. Through a more lightweight design, it can ensure the real-time performance of the network and improve the target detection ability.

### 2.2. YOLOv5 Network Structure

The YOLOv5 network is developed based on the previous-generation YOLOv3, YOLOv4, and other networks. Compared with the previous-generation networks, YOLOv5 has the characteristics of being faster and more accurate. For example, adaptive image scaling and an adaptive anchor box are used. These techniques obtain the scaling factor using the ratio of the current image size W and H and finally obtain the filled scaling size and effectively reduce the amount of calculation of the network. In terms of feature extraction, the cross stage partial (CSP) idea of YOLOv4 is mapped to the backbone network and neck layer of YOLOv5, which strengthens the ability of network feature fusion.

YOLOv5 has four network models, which are divided into s, m, l, and x from smallest to largest. The size mainly differs in the width and depth of the network. YOLOv5s is the lightest of them. The network is mainly composed of input, backbone, neck, and head. The input enriches datasets by using the Mosaic data enhancement module. The backbone uses the CSPDarknet53 backbone network to extract rich features from input images, including the focus module, spatial pyramid pooling (SPP) module, and so on to speed up network training. The neck core uses feature pyramid network (FPN) and path aggregation network (PAN) structures to fuse feature information at different scales. The bottom-up and top-down feature maps are laterally connected by the Concat operation, so as to realize the feature fusion of different deep and shallow scales. This improves the expressive power of the network.

The head is the detection structure of YOLOv5. Conv outputs three different sizes of feature maps: large, medium, and small, which correspond to the detection of small, medium, and large targets. YOLOv5 uses three loss functions to calculate the classification, confidence, and location losses and improves the accuracy of network prediction based on NMS. This study is based on the YOLOv5s network. The YOLOv5s network structure is shown in Figure 1.

## 3. Road Cracks

### 3.1. CBAM Attention Module

Because cracks account for a small proportion of the image, overlap with gray values of road materials, and are similar to the repaired crack shape, which directly results in lower recognition accuracy, the CBAM feature attention module [30] is introduced to improve the feature extraction ability of the network. The module internally contains a channel attention module and a spatial attention module, which consider the importance of pixels in different channels and different locations of the same channel in both spatial and channel dimensions to finally localize and identify the target, reduce the redundant information overwhelming the target due to convolution, and refine the extracted features. To avoid bias in network focus due to the premature inclusion of the attention mechanism, this module is added to the last layer in the backbone network. CBAM is shown in Figure 2.

The module operation process and the output can be expressed as (1) and (2), where *F* is the feature map of the input, $M_{c}$ is the channel attention mechanism, and $M_{s}$ is the spatial attention mechanism. ⊕ denotes the add operation, and ⊗ denotes element-wise multiplication. *F*″ denotes the final output feature map after the CBAM attention module. The output of the CBAM module is shown in Equations (1) and (2) as follows:(1)$$\begin{array}{rr} F^{\prime } & =M_{c}\left(F\right)⊗F \end{array}$$
(2)$$\begin{array}{rr} F^{\prime \prime } & =M_{s}\left(F^{\prime }\right)⊗F^{\prime } \end{array}$$

### 3.2. Bi-FPN Structure

In the feature fusion process, deep features and shallow features have different resolutions when fused across scales, and this difference directly affects the output results. The FPN-PAN structure of YOLOv5 uses an equivalent fusion strategy for input feature maps from different scales, while the Bi-FPN structure introduces adaptive weights in the feature fusion process at different scales, and adaptive weights can be gradually adjusted with deeper training, allowing the network to learn and enhance the differentiation ability of input feature importance, and to suppress or enhance different input features according to the weights, balancing the feature information between different scales. Its weighting formula is shown in Equation (3):(3)$$Out=\underset{i}{\sum }\frac{w_{i}\times fm_{i}}{\epsilon +\sum_{j}w_{j}}$$
where $w_{i}$ represents the learnable weights, and as the model is continuously trained, the value of this parameter changes with the optimizer update toward making the loss function optimal, and the value is set to 1 at initialization. $fm_{i}$ represents the input feature map in the network structure, ϵ is set constant equal to 0.0001 to ensure stable weight values, and the weight coefficients are normalized to between 0 and 1 using the ReLU function. For a layer in the middle of the network, the fusion is shown in Figure 3.

“⊕” denotes the add operation, and $w_{i}$ denotes the feature fusion weight values on different paths, where $w_{2}$ is the weight value on the directly connected path from $P_{4}^{in}$ to $P_{4}^{td}$, $w_{3}$ is the weight value on the cross-scale path from $P_{4}^{in}$, and $w_{4}$ is the weight value on the directly connected path from $P_{4}^{td}$. Finally, the process and output of feature fusion can be expressed as Equations (4) and (5) according to Equation (3):(4)$$P_{4}^{td}=Conv\left(\frac{w_{1}\times P_{4}^{in}+w_{2}\times Resize\left(P_{3}^{in}\right)}{w_{1}+w_{2}+\epsilon }\right)$$
(5)$$P_{4}^{out\;}=Conv\left(\frac{w_{3}\times P_{4}^{in}+w_{4}\times P_{4}^{td}+w_{5}\times Resize\left(P_{3}^{out\;}\right)}{w_{3}+w_{4}+w_{5}+\epsilon }\right)$$

To investigate the feature fusion effect of the Bi-FPN structure and whether the CBAM attention mechanism works, three sets of experiments were designed, and the same training techniques were used for the three sets of experiments, the batch size was set to 16, the epoch was set to 100 rounds, the initial learning rate was set to 0.01, and SGD algorithm was used. The experimental data are shown in Table 1.

After changing the network structure, the network parameters and FLOPS increased by 15.5% and 6.1%, respectively. mAP@0.5 increased by 1.8% and mAP@0.5:0.95 decreased by 2.2%. Experiments have shown that the CBAM module sends better feature maps to the neck layer, so Bi-FPN can better complete the cross-layer fusion for multi-scale features. The CBAM module insertion point is the last layer of the backbone and each feature fusion structure in the BIFPN structure. Due to the introduction of additional modules, the data read and write operations are increased, which raises the GPU computing cost and leads to a slight decrease in detection speed. In order to be able to meet the needs of high-precision and low-cost industrial tasks, this study continues to use depth-wise separable convolutional kernels to replace standard convolutional kernels to reduce the complexity of the model and improve the detection capability based on the current effect.

### 3.3. GS-BiFPN Structure

The GS-BiFPN structure is modified based on the Bi-FPN structure by replacing the original Conv module with GSconv and the C3 module with VoVGCSCP, which improves the feature fusion effect, speeds up the network inference, and effectively reduces the network complexity. The GSConv module is composed of a standard convolutional kernel, a depth-wise separable convolutional (DWConv) module, and a shuffle module [31]. The traditional DWConv module uses separate channels of convolution, which does significantly reduce the computational effort and the number of parameters, but also directly leads to the lack of feature information at the same spatial location during the convolution process and reduces the ability to extract features. In order to make up for this defect, the GSConv module combines the feature maps of the standard convolutional block and the DWConv module through the Concat operation and uses a shuffle strategy for the fused feature maps. The shuffle strategy mixes the feature information from the deep convolution module and the convolution module evenly, exchanges the feature information locally in such a way that the final feature map effect is as close as possible to the effect after standard convolution, and finally achieves a reduction in the number of parameters and FLOPs of the model while maintaining the accuracy. In addition, the VoVGCSCP module was designed based on the GSConv module, which reduces the complexity of the network. The structure of GSConv is shown in Figure 4, and the VoVGCSCP module is shown in Figure 5.

To verify the feature extraction ability of GSConv and how well it can optimize the parameters of the model, this study conducted experiments by adding the GSConv module to the backbone and neck, only the neck layer, and the original YOLOv5 model for comparison. In these three experiments, the input picture size was set as 640 × 640, batch size as 16, epoch as 100 rounds, initial learning rate as 0.01, and the SGD algorithm was adopted.

Comparing the three sets of experimental data, it can be seen that the feature extraction ability of GSConv is indeed inferior to that of the standard convolution kernel, and the drop in mAP@0.5:0.95 is obvious when it is applied to the backbone network. However, after the feature map has been effectively extracted by standard convolution in the backbone, the size of the feature map reaches the minimum value, and the number of channels reaches the maximum value when entering the neck layer. At this time, using depth-wise separable convolution has the minimum loss for feature extraction, ultimately achieving more effective extraction. The improvement of mAP@0.5 and mAP@0.5:0.95 values proves that the GSConv module effectively exchanged feature local information in the neck layer.

The reason why the number of parameters in group 1 and group 3 did not change significantly was that the width of the neck layer was much smaller than that of the backbone network, so the number of parameters did not decrease significantly.

To further improve the network detection ability, this study studied the feature extraction ability of the standard convolution module and the GSConv module in the neck layer. Group 1 using GSConv only in the neck layer and group 4 using GSConv for the overall network were added as a contrast. In the second and third groups, the standard convolution module is used to replace the GSConv module as the convolution module in the output small and medium object detection head. The same training technique was used in four sets of experiments, the batch size was set to 16, the epoch was set to 100 rounds, the initial learning rate was set to 0.01, and the SGD algorithm was used.

In the neck layer, the medium object detection head has a smaller size and a larger number of channels than the smallest object detection head. Comparing groups 1 and 3 of experimental data, the GSConv module can be compared with the standard convolution module in the feature extraction ability of deep networks. Compared with the experimental data of groups 1 and 2 of experiments, although the feature extraction of standard convolution is not as soft as GSConv in deep networks, the early use of standard convolution can provide high-quality feature maps with more obvious features for the later network, and improve the positioning ability of the network in the case of a high threshold. The results of the four groups of experimental data are basically in line with the research conclusions in Table 2. Considering these results comprehensively, this study uses group 2 as the improved network. In terms of detection speed, although GSconv does effectively reduce the complexity of the model, the introduction of modules such as depth-wise separable convolution increases the data processing process of the model, which directly leads to a reduction in detection speed. However, the FPS metric is still good for real-time detection tasks. In this study, the CBAM attention module and the improved GS-BiFPN feature fusion structure are introduced into the backbone to improve the low accuracy of the model. The improved GS-BiFPN structure is shown in the following Figure 6, where the pink square is the feature map after feature extraction.

The existence of the Bi-FPN provides a feasible approach for this study, but for the objective of this study, the performance based on the original Bi-FPN structure is not excellent, and the number of parameters and calculation amount do not meet the expected requirements of the study. Therefore, while retaining the idea of cross-scale connection, we made changes on the basis of the Bi-FPN structure to ease the problem of excessive resource consumption of the feature fusion structure, and we adopted the GSConv structure to make the feature fusion process softer, avoiding the excessive violent convolution operation of the original Bi-FPN structure resulting in the loss of target feature information. This makes the network focus too much on background information. Finally, according to Table 1 and Table 3, the number of network parameters and GFLOPS decreased by 4.7% and 11%, respectively. mAP@0.5 and mAP@0.5:0.95 increased 1.7% and 2.3%, respectively.

## 4. Traffic Signs

### 4.1. Four-Scale Detection Structure

In the complex road environment, traffic signs will be dense, and due to the scale transformation of the image during the driving process, the efficient detection of small targets is the focus of this research. The original YOLOv5 model uses three-scale feature layers for detection, and the scale sizes are 20 × 20, 40 × 40, and 80 × 80, respectively. It has a good detection effect on the COCO dataset with large targets, but for landmarks that are far away or small in volume, the target information will be lost in the convolution process, and the original three scales cannot complete the detection task with high accuracy. In view of the above problems, this study added a 160 × 160 large-scale detection head for small targets to the basic network structure of Yolov5. The network structure after the addition of a small target detection head is shown in Figure 7.

### 4.2. Data Augmentation

It is very effective to use data enhancement means for traffic signs. In real road conditions, there are many kinds of traffic signs, and their distribution is very random. Photometric distortion and geometric distortion are the most commonly used data enhancement methods, and data enhancement technology can directly and effectively improve robust network performance [32]. In the model of this study, Mixup [33], light distortion, and geometric distortion data enhancement methods were used based on Mosaic technology to help the network train the data. Mosaic refers to CutMix technology [34]. When Mosaic was applied, four samples were randomly selected for image transformation and Mosaic in different degrees before being sent to the training network, which increased the number of small targets and complicated the training samples. Mixup technology generates new data samples based on the proportional addition of two data samples, which provides continuous data samples for different types, expands the number of samples, and strengthens the learning ability in the network training stage.

## 5. Experimental Structure

### 5.1. Experimental Environment and Evaluation Index

In this experiment, the input picture size was set as 640 × 640, batch size as 16, epoch as 100 rounds, initial learning rate as 0.01, and the SGD algorithm was adopted. The experiment was carried out on a Windows 10 system with Intel(R) Xeon(R) Silver 4110 CPU @ 2.10 GHz, RTX 3070 Ti GPU, and 8 G memory.

Parameters, floating point operations per second (FLOPS), mean average precision (*mAP*), and FPS were used as the model evaluation indices in this study. TP means true positive. *FP* means false positive. *FN* means false negative. AP represents the average accuracy, $AP_{j}$ represents the average accuracy of the class j  target detection, c represents the category of markers, and represents the *mAP* mean of the average accuracy. AP@0.5j represents the average accuracy of class j targets when the intersection ratio threshold is 0.5, c represents the category of marks, $m_{AP@0.5}$ represents the average accuracy when the intersection ratio threshold is 0.5. $m_{AP@0.5:0.95}$ represents the mean value of mAP at the threshold of different intersection ratios with a step size of 0.05.
(6)$$R=\frac{TP}{TP+FN}$$
(7)$$P=\frac{TP}{TP+FP}$$
(8)$$AP=\int_{0}^{1}P\left(R\right)$$
(9)$$mAP=\frac{1}{c}\overset{c}{\underset{j=1}{\sum }}AP_{j}$$
(10)$$m_{AP@0.5}=\frac{\sum_{j=1}^{c}AP@0.5j}{c}\;\;\;$$
(11)$$m_{AP@0.5:0.95}=\frac{m_{AP@0.5}+m_{AP@0.55}+\ldots +m_{AP@0.95}}{10}$$

### 5.2. Dataset

In this study, the dataset was collected by Beijing High-Speed Transportation (First Group) in March and April 2022 using high-speed cruising cars equipped with fixed cameras on a section of a Beijing expressway. The cruising car was equipped with three cameras in front, rear, and roof. The type of camera in front and rear was DH-PTZ-33223-HNY-RB-B, and the type of camera in the roof was DH-IPC-HFW3233M-I1. The head-and-tail camera was responsible for recording road cracks; it was about 1050 mm from the ground and about 37 degrees from the horizontal line. The roof camera was responsible for recording traffic signs; it was about 1540 mm from the ground and at an angle of about 22 degrees from the horizontal line. The data were collected along the high-speed road at an average speed of 80 KM/H. Finally, 2164 1920 × 1080 pixel original road ground images were collected for the production of the road crack dataset, and 8146 1920 × 1080 pixel original traffic sign images were collected for the production of the road traffic dataset. The dataset was made using labeling. Both datasets were divided according to the ratio of 80% as the training set and 20% as the verification set.

There were three types of targets in the road Crack dataset, namely Crack, Rcrack (Repaired crack), and Expansion Joint. The proportions are shown in Figure 8a. A total of 18 categories were marked in the traffic sign dataset, namely HLF1 (Path Gantry), HLF2 (Speed Gantry), HLF3 (Monitor Gantry), HLF4 (LED Guidance Screen Traffic Gantry), HLF1Camera (Path Gantry Camera), HLF4Camera (LED Guidance Screen Traffic Gantry Camera), GreenRS (Green Traffic Sign), BlueRS (Blue Traffic Sign), WhiteRS (White Traffic Sign), BrownRS (Brown Traffic Sign), YellowRS (Yellow Traffic Sign), Camera1 (SkyNet Surveillance Camera), Camera2 (Traffic Flow Monitoring Camera), Camera3 (Bayonet Camera), CircleCamera, and Camera4 (Violation Capture Camera). This basically covers the main traffic environment on expressways, in which small targets account for about 34.8% of the traffic sign dataset. The proportions of all types and targets of different sizes are shown in Figure 8b,c.

### 5.3. Experiment and Analysis

In order to verify the performance of the improved model, this study compares and analyzes the current mainstream one-stage object detection models, with each model using the same training technique. The experimental results are shown in Table 4. In the two indicators, mAP@0.5 and mAP@0.5:0.95, the method used in this study is significantly better than other detection models. The number of parameters in the training process is basically the same as that of the original YOLOv5 network, and the number of FLOPS is slightly higher than that of YOLOv3-tiny and YOLOv7-tiny-silu. In the validation process, the method used in this study was able to achieve 58 FPS. In general, the improved YOLOv5 model can achieve high detection ability while remaining lightweight, and it can play a better role than most models in low-cost industrial detection tasks. In the following experiments, the batch size was uniformly set to 16 and the epoch was uniformly set to 100.

In order to reflect the good detection ability of the proposed method and explore the informativeness of each improved method, this study designed a total of four groups of ablation experiments, and the training methods and hyperparameter values used in the four groups of experiments were the same. The experimental results are shown in Table 5, where “√” indicates that the module is introduced and “×” indicates that the module is not used. In the following experiments, the batch size was uniformly set to 16 and the epoch was uniformly set to 100.

It can be seen from Table 5 that the YOLOv5 model after the integration of the CBAM attention mechanism and GS-BiFPN feature fusion network has a significant improvement compared with the original YOLOv5s, with mAP@0.5 and mAP@0.5:0.95 increased by 3.5% and 0.1% respectively. Finally, the method proposed in this study is visualized in Figure 9.

The same comparison experiment was carried out for traffic signs, and the same training technique was used for each model. The experimental results are shown in Table 6. Since road markings are far more complex than road cracks, the experiment is evaluated for small, medium, and large targets, where APS, APM, and APL are the mean values of mAP at different intersection ratio thresholds of IOU = 0.5:0.95. It can be seen from the table that the model in this study has good performance on small targets and large targets and is better than other mainstream models. The test power for medium-sized targets is only 0.4% lower than that of the TPH-YOLOv5 model. The number of parameters is basically the same as that of YOLOv5, and the number of FLOPS is slightly higher than that of YOLOv5. In the verification process, the method used in this study was able to achieve 52 FPS with a GPU. The overall performance shows that the improved model can be competent for low-cost industrial detection tasks. In the following experiments, the batch size was uniformly set to 16 and the epoch was uniformly set to 100.

In order to reflect the good detection ability of the proposed method and explore the informativeness of each improved method, this study designed four groups of ablation experiments for the traffic sign dataset, and the training methods and hyperparameter values used in the four groups of experiments were the same. The experimental results are shown in Table 7. In the following experiments, the batch size was uniformly set to 16 and the epoch was uniformly set to 100.

As can be seen from Table 7, in the four-scale detection structure, the data for large and small targets are improved, and the addition of large-scale detection heads indeed effectively captures the targets in the shallow network. After integrating the four-scale detection structure and data enhancement methods, the YOLOv5 model is significantly improved compared with the original YOLOv5s. The detection accuracy for large and medium-sized targets is improved by a small margin, and the detection accuracy for small targets is increased by 12.2%, which effectively shows that the improved model has a high ability to detect small targets. Its visualization is shown in Figure 10.

## 6. Conclusions

In this study, road cracks and traffic signs were studied separately, and then two improved YOLOv5 real-time detection networks were proposed. For road cracks, this study adds a CBAM attention module based on the original YOLOv5 model and uses the proposed GS-BiFPN structure to replace the feature fusion structure, which balances the feature information between different scales and improves the expressive ability of the network. Experiments show that on the self-made road crack dataset, the average accuracy of the improved algorithm can reach 69.9%, which is 3.5% higher than that of the original model. The detection speed can reach 58. For traffic signs, this study replans the network structure and finally designs a four-scale feature detection structure, and it combines the current mainstream data enhancement methods for the training phase to strengthen the network learning ability. The experimental results show that the average accuracy of the improved network reaches 63.0% in the self-made traffic sign dataset trained under the high precision threshold, and the average accuracy for the small target reaches 43.6%, which is 12.2% higher than that of the original network. The detection speed can reach 52. The performance of the two improved networks in the corresponding dataset is better than that of the current mainstream one-stage detection network, and the trained models are very lightweight and suitable for mobile deployment. The overall performance shows that the improved networks can be competent for low-cost real-time industrial detection tasks.

However, the classification task of the two datasets is not perfect. In the actual application scenario, the classification of road cracks and traffic signs is more complex. In the future, it will be necessary to improve the database and improve more kinds of classification recognition. It is also hoped to develop a road asset classification system with a complete interface and a soft and hard platform, which is more conducive to the overhaul and maintenance of road inspection staff.

## Figures and Tables

**Figure 1 sensors-23-04589-f001:**
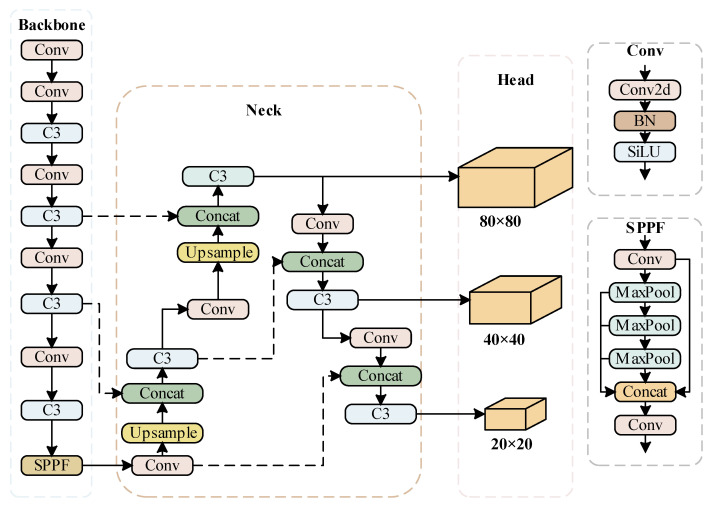
Algorithm Structure Diagram of YOLOv5.

**Figure 2 sensors-23-04589-f002:**
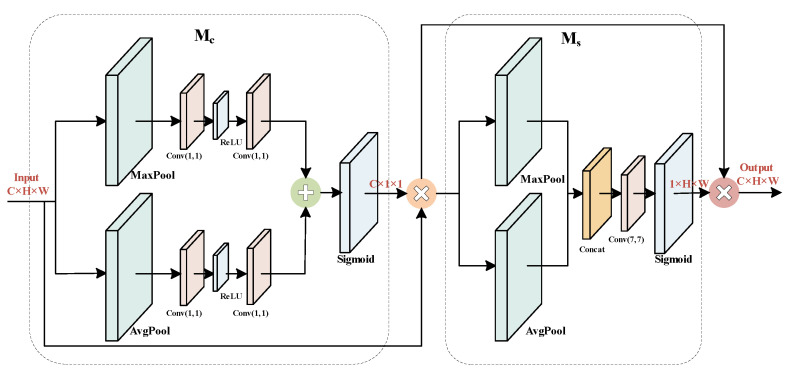
Structure of CBAM attention module.

**Figure 3 sensors-23-04589-f003:**
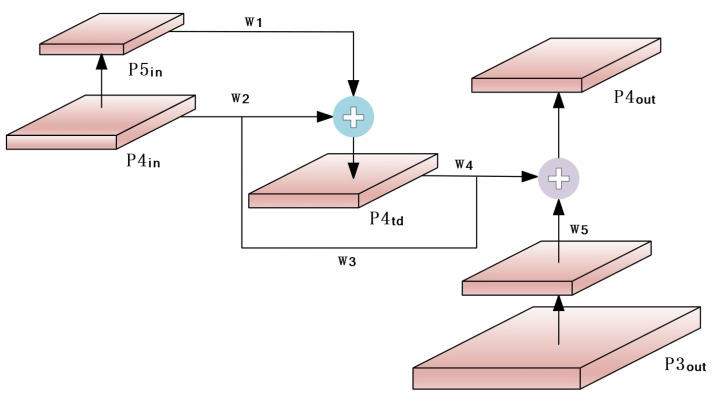
Cross-Layer Convergence Architecture Diagram.

**Figure 4 sensors-23-04589-f004:**
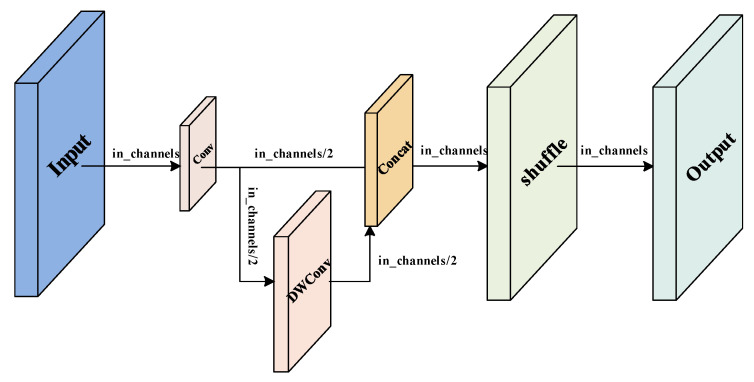
Structure of GSConv.

**Figure 5 sensors-23-04589-f005:**
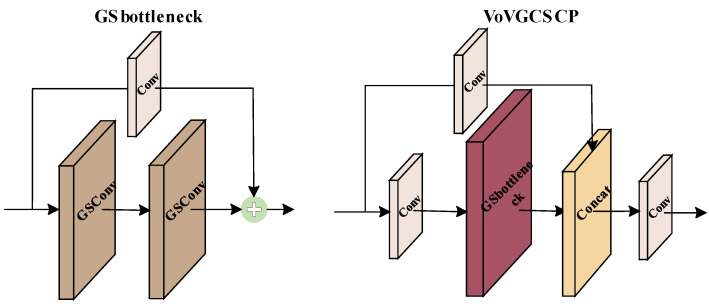
Structure of VoVGCSCP.

**Figure 6 sensors-23-04589-f006:**
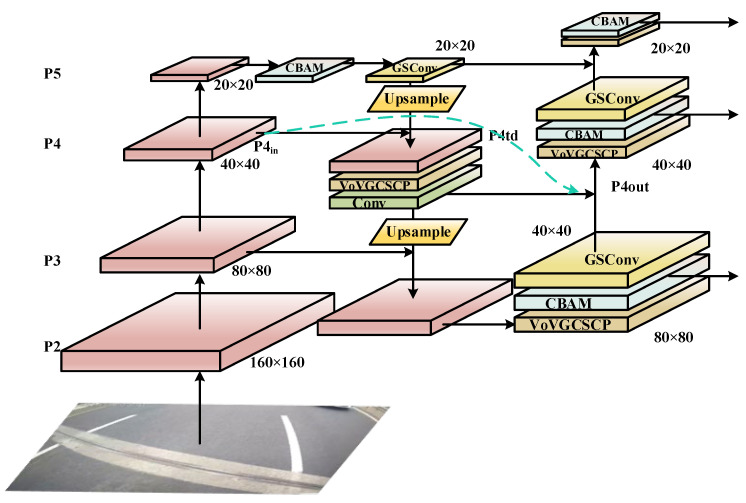
Structure of GS-BiFPN.

**Figure 7 sensors-23-04589-f007:**
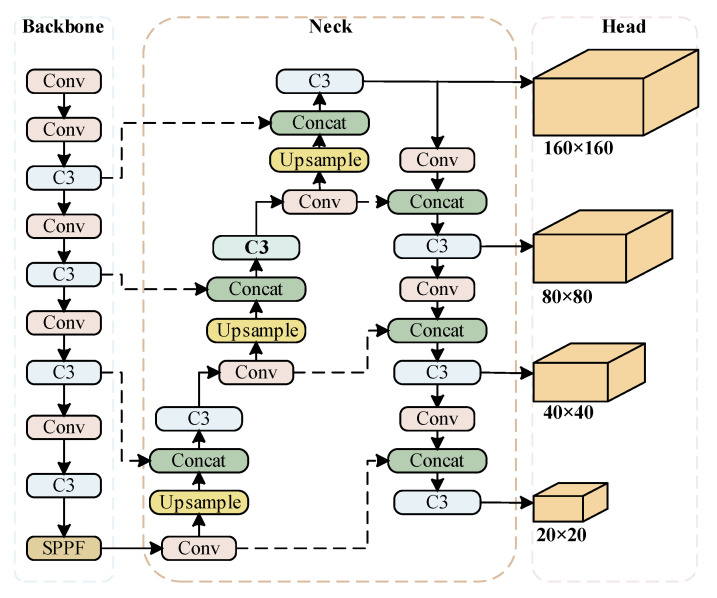
Improved structure of multi-scale detection.

**Figure 8 sensors-23-04589-f008:**
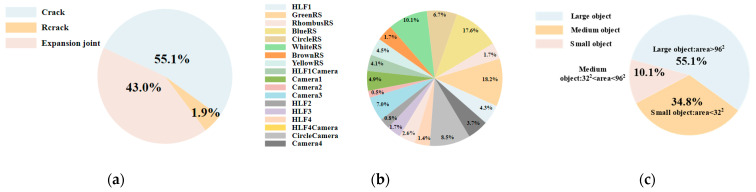
Figure of the proportions of categories in each dataset. (**a**) Size distribution of crack instances from the crack dataset. (**b**) Map of sign instances. (**c**) Size distribution of sign instances from the sign dataset.

**Figure 9 sensors-23-04589-f009:**
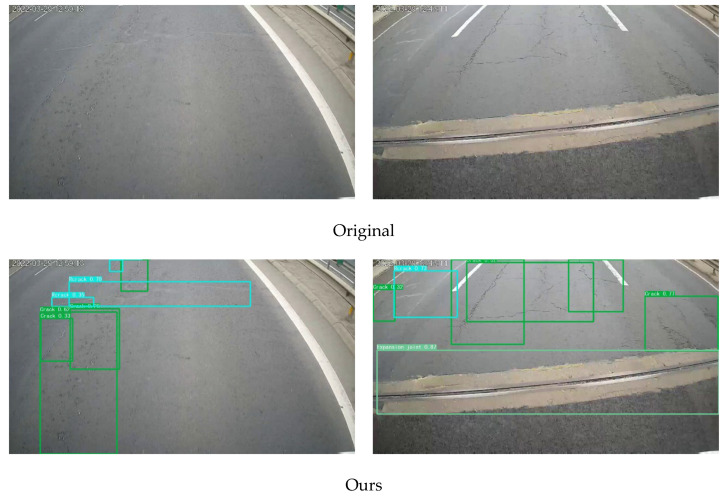
Some examples detected by our method in the road crack dataset.

**Figure 10 sensors-23-04589-f010:**
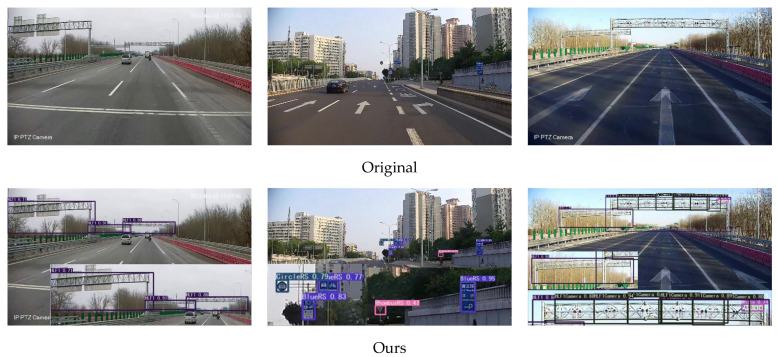
Some examples detected by our method in the traffic sign dataset.

**Table 1 sensors-23-04589-t001:** Bi-FPN, CBAM contrast experiments.

Model	Parameters	FLOPS	mAP@0.5/%	mAP@0.5:0.95/%	FPS
YOLOv5s	7.1 M	16.3 G	66.4	36.5	78
YOLOv5s+bifpn	8.1 M	17.3 G	67.3	32.9	63
YOLOv5s+cbam+bifpn	8.2 M	17.3 G	68.2	34.3	59

**Table 2 sensors-23-04589-t002:** GSConv module feature extraction ability contrast experiments.

Model	Parameters/10^6^	GFLOPS	mAP@0.5/%	mAP@0.5:0.95/%	FPS
YOLOv5	7.1	16.3	66.4	36.5	78
Backbone + neck	6.7	11.6	66.3	34.4	53
Only neck	7.1	14.9	68.3	36.3	61

**Table 3 sensors-23-04589-t003:** GSConv and standard convolution feature extraction ability in neck layer contrast experiments.

Model	Parameters/10^6^	GFLOPS	mAP@0.5/%	mAP@0.5:0.95/%	FPS
Neck-GSConv	7.9	15.4	70.6	34.6	55
Small-head	7.9	15.4	69.9	36.6	58
Medium-head	8.0	16.0	70.6	34.3	55
All-GSConv	7.4	12.3	65.7	33.5	50

**Table 4 sensors-23-04589-t004:** Road crack contrast experiments.

Method	Parameters	FLOPS	mAP@0.5/%	mAP@0.5:0.95/%	FPS
YOLOv5s	7.1 M	16.3 G	66.4	36.5	78
YOLOv7-tiny-silu [35]	6.2 M	13.2 G	67.9	30.7	**101**
TPH-YOLOv5 [36]	9.2 M	23.0 G	66.0	34.4	27
YOLOv3-tiny	8.7 M	12.9 G	44.1	13.4	95
Mobilenet-SSD-Lite [37]	25.1 M	29.2 G	59.9	25.0	60
Efficientdet-d0	3.8 M	2.5 B	62.0	27.9	17
**Ours**	7.9 M	15.4 G	**69.9**	**36.6**	**58**

**Table 5 sensors-23-04589-t005:** Road crack ablation experiments.

Group	CBAM	GS-BiFPN	Crack	Rcrack	Expansion Joint	mAP@0.5/%	mAP@0.5:0.95/%
1	×	×	51.4/21.2	58.4/25.0	89.5/63.4	66.4	36.5
2	√	×	46.5/19.7	62.0/28.9	89.1/49.9	65.9	32.9
3	×	√	49.9/20.3	63.9/29.0	89.7/52.8	67.8	34.0
4	√	√	52.6/21.6	63.4/28.2	93.8/60.0	69.9	36.6

**Table 6 sensors-23-04589-t006:** Traffic sign contrast experiments.

Method	Parameters	FLOPS	AP_S_/%	AP_M_/%	AP_L_/%	mAP@0.5:0.95/%	FPS
YOLOv5s	7.1 M	15.9 G	31.4	49.0	59.3	53.3	77
YOLOv7-tiny-silu [35]	6.1 M	13.2 G	27.4	51.5	61.0	56.3	**112**
TPH-YOLOv5 [36]	8.4 M	22.7 G	39.5	**59.2**	66.3	**63.2**	33
YOLOv3-tiny	8.7 M	12.9 G	21.2	46.9	56.2	50.1	88
Mobilenet-SSD-Lite [37]	25.1 M	29.2 G	4.6	17.0	41.8	28.4	39
Efficientdet-d0	3.8 M	2.5 B	0.9	36.6	63.2	45.3	17
**Ours**	7.7 M	27.3 G	**43.6**	**58.8**	**67.1**	**63.0**	**52**

**Table 7 sensors-23-04589-t007:** Traffic sign ablation experiments.

Method	AP_S_	AP_M_	AP_L_	mAP@0.5/%	mAP@0.5:0.95/%
YOLOv5s	31.4	49.0	59.3	86.4	53.3
YOLOv5s-aug	38.0	**58.9**	66.4	**92.7**	62.8
YOLOv5s-Fourdetect	33.8	52.4	62.4	87.1	57.2
**Ours**	**43.6**	**58.8**	**67.1**	**92.4**	**63.0**

## Data Availability

The data presented in this study are available on request from the corresponding author.

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
