# Peer review of "Improved YOLOv5 Network for Real-Time Object Detection in Vehicle-Mounted Camera Capture Scenarios"

_sensors, 2023, doi:10.3390/s23104589_

Round 1

Reviewer 1 Report

The author presents in this paper improved YOLOv5 network for real-time object detection in vehicle mounted camera capture scenarios. The manuscript is generally well-designed and the results are adequate from a scientific point of view. However, the paper can be accepted after the following modifications.

1.   Related Work title needed to be in next page, the same about table 3 title, table7.

2.   literature review is just a pile of information, lacking of analysis and induction

3.   We would like to see a diagram showing the proposed work. This helps to simplify and improve the structure of the text.

4.   We suggest adding the link of the dataset for giving an opportunity to the scientific community to access the data easily.

5.   Please add more information about training, validation (if exist) and test set of data set used.

6.   The authors should have explained the choice of hyperparameters in proposed model such as number of epoch, size of batches, etc.

7.   Are authors used the images include complex and changeable conditions in experimental study? If yes, they need to add some results for each conditions.

8.   The authors should have employed cross-validation of improved yolo for the evaluation method.

Reviewer 2 Report

This paper proposes two improved YOLOv5 object detection networks to detect road cracks and traffic signs. While this research topic is of interest to the civil and transportation community, the following comments need to be addressed properly:

Major comments:

1.       Multi-scale feature fusion network through Bi-FPN is an existing technique. The manuscript needs to distinguish between the existing Bi-FPN technique and the proposed feature fusion module in this manuscript and emphasize its difference/novelty.

2.       This manuscript actually focuses on two different topics, namely the detection of road cracks and traffic signs. The narrative of this manuscript is thus separated into two branches, which, however, are not closely integrated. Please consider further elaborate the differences between the proposed two networks in dealing with cracks vs. traffic signs.

3.       Under what weather were the road crack and traffic sign datasets collected? And what, if any, are the influences from the weather impact? Crack detection using RGB image is often subject to performance deterioration due to environmental factors such as changing illumination conditions. This paper is focused on real-time application in practical scenarios, and therefore the weather impact definitely needs to be taken into consideration.

4.       As shown by the crack detection results in Figure 11, some identified crack objects overlapped with each other. How does the proposed algorithm consider the overlapped detection during performance evaluation? And what if multiple adjacent crack objects are detected as one single crack object by the algorithm?

5.       Further details need to be provided regarding the instrumentation of the cameras mounted on the survey vehicle, such as the camera height above ground, view angle, focal length, etc.

6.       In addition to detecting cracks and repaired cracks, the reviewer suggests modifying the crack detection module to further classify the crack objects into several subcategories, such as longitudinal crack, transverse crack, block crack, alligator crack, etc. Besides, other roadway defects such as potholes may be considered the network output as well. This is a suggestion for future work and the authors do not have to implement it into the manuscript.

Editing:

1.       Page 2, remove the redundant descriptions on the “summary to the contributions to road cracks”

Overall, the quality of English is good.

Reviewer 3 Report

1. The sections of the paper must be clear. There is is need for the introduction, study areas, methodolgy, results & discussion and the conclusion. The authors are mixing information at the moment and there i need to separate literature, what was done and the results. 

2. Some of the information that coming from literature is not referenced and this must be addressed. 

3. The paper was written in third person and in some instances the authors were used first person reporting (we/our/us etc). There is need for them to be consistent. 

4. There are too many abrevations and these must be written in full on first mention. 

5. On the Yolov5 the authors must give more information about the older version and in the methodolody they must include what they did to improve the Yolov5 network. 

These are my observation and I hope they will help improve the this research paper. 

Regards

James
